# People choose to receive human empathy despite rating AI empathy higher
Joshua D. Wenger [1] ✉, C. Daryl Cameron[1,2] & Michael Inzlicht[3,4]

Recent advances in AI have enabled large language models to produce expressions that seem empathetic to human users, raising scientific and ethical questions about how people perceive and choose between human and AI sources of emotional support. Although an increasing number of studies have examined how people rate empathy generated by AI, little to no work has examined whether people would choose to receive empathy from AI. We conducted four studies investigating whether people prefer to receive empathetic expressions from humans or AI, and how they evaluate these expressions. Across diverse samples and stimuli, we found evidence for what we term the AI empathy choice paradox: participants significantly preferred to receive empathy from humans, yet they rated AI-generated empathetic responses as higher in quality, more effective at making them feel heard, and more effortful when they did choose them. These findings contribute to ongoing debates about AI empathy by demonstrating that while people may avoid AI as an empathy source, they nonetheless benefit from AI empathy when they experience it. Our results suggest potential applications for AI in supplementing human emotional support while highlighting the importance of respecting individual preferences for empathy sources.

## People choose to receive human empathy despite rating AI empathy higher

In an era where artificial intelligence (AI) has redefined technological boundaries, large language models like ChatGPT now generate outputs that simulate deeply human capacities, including empathy and emotional understanding. While AI applications in coding, healthcare, and data analysis have gained acceptance[1,2] the emergence of AI in emotional and relational domains has sparked intense debate among researchers and ethicists alike[3,4]. This proliferation of seemingly empathetic AI interactions presents a provocative question: Do people want or benefit from empathy expressed by entities incapable of genuine emotional experience?

As millions now regularly encounter AI systems designed to recognize and respond to human emotional states, understanding how people navigate these novel empathic exchanges has become increasingly urgent. We explore individuals' preferences to receive empathetic responses from human versus AI sources, and analyze how each source affects emotional experience, revealing a striking paradox that challenges assumptions about technology and human connection.

The study of empathy from AI is controversial in itself. Empathy includes multiple components, such as experience sharing, which refers to sharing in the experiences and feelings of the empathy recipient; perspective taking, the act of mentally representing and understanding what the recipient is thinking or feeling; and empathic concern or compassion, the emotional response to the recipient's experience and the motivation to reduce their suffering[5]. However, because AI lacks emotional experience, it is incapable of sharing in the experience of the empathy recipient and has no genuine motivation to alleviate their suffering[6]. Therefore, although AI expressions of empathy may seem authentic, they are, in reality, fake and, some have argued, deceptive[7]. This lack of authentic empathic capacity has led many to be skeptical of its use in the empathy domain[6–8].

Others, however, have argued against the notion that this distinction between "fake" and "true" empathy is helpful[9,10]. From a recipient-focused perspective on empathy, what may matter more is how a recipient feels and what needs and motives are satisfied through receiving empathic support, regardless of source. According to this perspective, AI empathy offers several potential advantages over human empathy, including resistance to empathy avoidance and fatigue, broad accessibility to users regardless of location or resources, and consistent reliability in quality of emotional support[9,10]. Therefore, while acknowledging AI's lack of genuine empathic capacity, there is also a reasonable argument for deemphasizing the question of "real-

[1]Psychology Department, The Pennsylvania State University, University Park, PA, USA. [2]Rock Ethics Institute, The Pennsylvania State University, University Park, PA, USA. [3]Psychology Department, The University of Toronto, Toronto, ON, Canada. [4]Rotman School of Management, The University of Toronto, Toronto, ON, Canada. ✉e-mail: jdw6278@psu.edu

ness" and focusing on the question: Does the individual receiving expressions of empathy from AI feel cared for, comforted, validated, etc.?

How do people actually feel when receiving empathy from AI? One study examined the potential value of AI responses within the healthcare domain[1]. The researchers scraped a variety of healthcare questions from users on the subreddit r/AskDocs, a forum where people post medical questions for verified physicians to answer. For the collection of healthcare questions, researchers scraped physician responses, and used AI to generate additional responses. These responses were then rated by medical professionals who were blind to whether AI or other doctors created the responses. The AI responses were preferred in 78.6% of the 585 cases. Critically, the AI responses were not only judged as superior in quality, but as more empathetic.

Another study found that AI responses were rated by third-person observers as more compassionate than a curated set of human responses selected on the basis of being rated as high quality[11]. This held for both negative and positive situations. These findings remained regardless of whether the responses were labeled as human or AI. Even more impressively, the superior compassion ratings of AI responses held even when compared to human responses generated by trained human empathizers (crisis responders thought to be "compassion experts"). These results built upon the previous work[1], expanding beyond a medical context to everyday social decisions with more tightly controlled stimuli.

Lastly, a more recent set of studies examined empathy from the recipient's perspective rather than third-party evaluations, similarly finding that recipients rated AI responses as higher in quality[12]. In this work, participants described a "complex situation they were dealing with" (p. 2) and were randomly assigned to receive either a human or AI empathetic response. The participants then rated their received response (tailored to their situation) on various metrics such as feeling heard, response accuracy, feeling understood, and connection to the responder. Responses that were generated by AI were rated as superior to human-generated responses on all metrics. These responses, however, also varied in whether they were labeled as coming from a human or AI. Responses labeled as coming from human versus AI empathizers were rated higher in empathetic quality, revealing an empathic penalty for the AI label.

This so-called label effect has since been replicated in another study in which all participants were given AI-generated empathy responses which were randomly labeled as authored by a human or AI[13]. Responses labeled as human-authored were rated as more empathetic. The researchers also asked participants to identify the degree to which they believed the human responses were edited by AI, and they found that human-labeled responses were perceived as less empathetic the more they believed the response to be AI-edited. In this way, there is a push and pull to AI empathy such that: 1) AI responses tend to be rated as higher in empathetic quality, yet 2) the AI label somewhat diminishes their impact.

While this previous research has consistently demonstrated that AI empathy is rated as superior in quality on several metrics (e.g., compassion, feeling heard), most of this work has *assigned* people to receive empathy from a human or AI, rather than allowing people to *choose* where they seek empathy from, as in real-world contexts. Normative discussions about what people would and should prefer[8,14] depend on a clear understanding of what people descriptively prefer when given the opportunity, which has received little empirical test. Regardless of the advantages AI empathy expressions may have relative to human empathy, we need to consider individuals' actual preferences because the advantages are a moot point if people do not seek out AI in the first place. Here, we use a choice paradigm to explore exactly this question: Are people willing to seek out empathy from AI?

## Methods
### Study 1
The initial study sought to answer our primary question of interest: Do people prefer to choose a caring response from humans or AI? This study intended to establish a baseline estimate of choice preferences. We additionally explored the question: When participants choose to receive a caring

response from an AI, how does this AI response make them feel relative to the responses they receive when choosing a human response?

The initial sample included 153 participants recruited from Prolific and screened for English fluency using an inbuilt Prolific pre-screener. The sample size was calculated using G*Power 3.1 to detect a small-medium effect for a paired samples and one sample t-test with 80% power[15]. One participant was excluded from the analyses due to copying and pasting fragments of the stimuli across free response questions, leaving a final sample of 152 participants (95 female, 53 male, 4 nonbinary; $M_{age}$ = 37.02 years, $SD_{age}$ = 12.37; 95 White/Caucasian, 23 Black/African American, 10 Hispanic/Latino, 7 Asian/Pacific Islander, 1 Native American, 1 other, 15 mixed).

Building on motivated empathy perspectives[16–18], we used a modified version of the empathy selection task[19]. This task views empathy as a motivated process in which individuals weigh its costs and benefits to ultimately make a decision to empathize. We adapted this general approach to examine motivated empathy *reception* rather than *expression*. Instead of presenting participants with situations in which they choose whether to *express* empathy for a target, participants could make the choice of which source to *receive* empathy from: human or AI.

In this task, participants were informed that they would repeatedly be presented with two decks of cards labeled "HUMAN" and "AI" and that they should freely choose between them. Depending on which block they were completing, the instructions said they would either, "receive an empathetic response in which the responder will share in your experience" (empathy block), or, "receive a compassionate response in which the responder will express warm feelings of concern" (compassion block), created by the source of whichever deck they chose.

We emphasize that, in this design, participants received responses that were actually created by the source of their choosing. That is, if they selected the "HUMAN" deck, they received a human-created response and if they selected the "AI" deck, they received an AI-created response. Other studies have experimented with dissociating the true response author from the author label (e.g., labeling an AI-created response as human)[12,13]. These studies typically find that people penalize AI sources of empathy, holding content constant. In contrast, we are interested in whether people willingly prioritize empathy reception from humans or AI, while *fully knowing* the source. That is, we do not simply explore the separate effects of actual response authorship crossed with author label, but instead explore the unique question about empathy choice and the emotional consequences thereof, given accurate information of authorship. We believe this question is closer to real life empathy-seeking decisions as, naturalistically, people are likely to know that they are interacting with a human or a chatbot.

Human responses were generated by undergraduate research assistants and AI responses were generated by ChatGPT-4o[20]. Both response generators were given a list of 20 vignettes depicting various types of suffering (half depicted physical suffering and half emotional suffering) adapted from prior work[21]. For each vignette, the undergraduate RAs and ChatGPT were instructed to:

"Imagine a person in the following scenario. Write two responses to the person in this situation:

1) A compassionate response in which you generate warm feelings of concern for the person
2) An empathetic response in which you share in the experience of the person"

We aimed to have both human and AI respond using instructions adapted from prior iterations of the empathy selection task that assessed choices to experience share[19] and to cultivate compassion[22]. We differentiate between empathy and compassion as an exploratory factor in the design, but we analyze in the supplemental materials (pp. 15–23).

Five undergraduate research assistants each crafted responses for all 20 vignettes, one response for compassion and one for empathy, totaling 200 unique human responses. By using a pool of human responses, we hoped to more accurately represent the variability that exists within human empathy,

rather than attempting to generalize to human empathy at large while using stimuli from a single empathizer's idiosyncratic form of empathy. Similarly, due to the stochastic/random nature of ChatGPT, AI responses were created by prompting ChatGPT five separate times for each vignette. In total, we generated 20 unique responses for all 20 vignettes (five AI compassion responses, five AI empathy responses, five human compassion responses, and five human empathy responses per vignette), all of which are available in the OSF repository.

Participants completed a total of 20 trials; 10 trials in the empathy block where they received empathetic responses from a human or AI (based on their choice in each individual trial), and 10 trials in the compassion block where they received compassionate responses from a human or AI. All participants saw all 20 vignettes, but they were separated into two lists of 10 vignettes each and counterbalanced such that half of participants saw the List A vignettes in the empathy block and List B in the compassion block, and the other half saw the opposite pairing (List A vignettes in the compassion block and List B vignettes in the empathy block). Each list consisted of five vignettes depicting emotional suffering and five vignettes depicting physical suffering. The lists were determined by the research team with the assistance of ChatGPT to be equivalent in terms of suffering severity and type. We additionally counterbalanced block order across participants. Similar to empathy vs. compassion, we treat physical vs. emotional suffering as an exploratory factor and report analyses in the supplemental materials rather than main text (pp. 15–23).

At the beginning of each trial, participants were presented with a vignette depicting a negative situation (see the OSF repository for the full vignette list[21]). For example:

"You were going to visit your father. There was so much that you had never told your father, and you finally wanted to let him know how much you cared. When you got to your parents' house an ambulance was there. Your father had died in his sleep."

Participants were told to visually imagine themselves in the situation and write three words which described their emotional experience (a question included to increase immersion in the vignette). They were then presented with a choice between the "HUMAN" and "AI" decks. Following this choice, participants received a response that was pre-generated by the source of their chosen deck. The response was randomly selected from the pool of five responses corresponding to the specific vignette and their specific choice of human or AI. That is, if participants chose "HUMAN," they randomly received a response pre-generated by one of the five undergraduate research assistants that corresponded to that specific vignette, and the specific block (empathy or compassion) that the participant was completing. Participants then rated the response quality in terms of their agreement (seven scale anchors ranging from "Strongly Disagree" to "Strongly Agree") with the following statements: "The response was empathetic", "The response was compassionate", "I feel connected to the responder", "Reading this response makes me feel comforted", "Reading this response makes me feel cared for", "Reading this response makes me feel validated." These same six rating questions were included after each trial, regardless of block. In addition to these trial-level ratings, participants indicated how much effort the responders put into their responses from each respective deck (on a five-point scale, 1 = very low, 5 = very high). They made these effort ratings for the "HUMAN" and "AI" decks separately after each block rather than trial in order to reduce overall survey time and participant burden. Details about additional measures can be found in the supplemental materials (p. 2). As described below, we used multilevel modeling and an aggregated one-sample t-test as our primary analytic methods.

## Study 2
In this study we sought to replicate and extend Study 1 in a sample of undergraduate college students. Based on the results from Study 1, we pre-registered two hypotheses:

H1: Participants would choose to receive caring responses from humans more than AI.

H2: Participants would rate AI responses more favorably than human responses.

The initial sample included 201 participants recruited from an undergraduate research pool at a large public university in the United States. The sample size was calculated using G*Power 3.1 to detect a small-medium effect for a paired samples and one sample t-test with 99% power. Five participants were excluded from the analyses based on our pre-registered exclusion criteria, leaving a final sample of 196 participants (153 female, 43 male; $M_{age}$ = 18.92 years, $SD_{age}$ = 1.31; 134 White/Caucasian, 9 Black/African American, 12 Hispanic/Latino, 16 Asian/Pacific Islander, 4 other, 21 mixed).

Study 2 followed the same general experimental structure as Study 1, with a slight differences. After every trial, participants indicated their level of agreement with the following six statements: "The response was empathetic", "The response was compassionate", "The response was caring", "Reading this response makes me feel comforted", "Reading this response makes me feel cared for", "Reading this response makes me feel validated."

We also included the following two exploratory ratings after each trial: "I feel connected to the responder" and "The response felt authentic." Connectedness to responder and authenticity are two constructs that have been examined in the AI empathy literature previously[12,23]. We believed these ratings could be sensitive to whether participants think of AI empathizers and its empathy as fake, and could be areas in which humans may have an advantage. As an additional exploratory predictor, participants also indicated their suffering intensity after picturing themselves in each vignette, but before making their choice ("How much suffering are you experiencing" 0 [none at all] - 10 [extreme]). They additionally answered the previous block-level question regarding the perceived effort of the responder after each trial. We used multilevel modeling and an aggregated one-sample t-test as our primary analytic methods. Analyses of exploratory ratings, exploratory factors (empathy vs. compassion and physical vs. emotional suffering), and details about additional measures can be found in the supplemental materials (pp. 4 and 23-31).

## Study 3
In our next study, we sought to replicate and extend Study 2 using an alternative stimulus set from prior work that was normed to be high-quality human empathy[11]. In the supplemental materials (pp. 31–34), we also explored whether scenario valence matters, given that empathizing with positive experiences tends to be common within everyday relationships[24]. We pre-registered the following hypotheses:

H1: Participants would choose to receive caring responses from humans more than AI.

H2: Participants would rate AI responses more favorably than human responses.

H3: There would be an interaction between response source and valence such that the difference in response rating would be even greater for trials depicting negative scenarios relative to positive (building on prior work[11]; explored in the supplement, p. 33).

The initial sample included 200 participants recruited from an undergraduate research pool at a large public university in the United States. The sample size was calculated using G*Power 3.1 to detect a small-medium effect for a paired samples and one sample t-test with 99% power. Five participants were excluded from the analyses based on our pre-registered exclusion criteria, leaving a final sample of 195 participants (151 female, 41 male, 2 nonbinary, 1 preferred not to answer; $M_{age}$ = 18.91 years, $SD_{age}$ = 1.64; 128 White/Caucasian, 10 Black/African American, 16 Hispanic/Latino, 12 Asian/Pacific Islander, 1 other, 1 non-response, 27 mixed).

Study 3 followed a similar experimental structure to the previous studies, the key difference being the use of this alternative stimulus set[11]. We adapted these vignettes to use second-person rather than third-person language, as second-person language is what fits most naturally with our experimental paradigm in which participants are instructed to imagine themselves in the presented situations. This stimulus set consisted of a

different set of vignettes, a different pool of AI responses, and a different pool of curated, high quality human responses. That is, unlike our previous human responses which represented average human empathy, this set contained a mix of human responses either created by Prolific users and rated as highly empathetic by graduate students and undergraduate research assistants, or created by trained empathizers (crisis responders). These responses and vignettes did not distinguish between empathy and compassion or physical and emotional suffering, but instead depicted a mix of negative and positive events. As such, participants completed only a single block of 16 trials, half of which contained a negative vignette and half of which contained a positive vignette. The general trial structure and measures were identical to Study 2, with the exception of the removal of the question regarding suffering intensity (as half of the trials did not depict suffering). We used multilevel modeling and an aggregated one-sample t-test as our primary analytic methods.

## Study 4

In our final study, we sought to expand on a limitation of the prior studies by exploring empathy choice for participants' own real-life scenarios, rather than imagined scenarios with pre-generated responses. That is, we examined whether results also held when participants made empathy-seeking decisions and received responses for their own personal experiences for which they desired empathy. The previous studies used vignettes and pre-generated responses because they allowed us to have participants complete a larger number of choice trials (i.e., it would be less feasible to ask participants to recount 20 experiences from their own life for which they would like empathy), yielding a highly reliable estimate of choice preference. Additionally, these designs made it possible to provide empathy responses immediately following participant choices. However, the tradeoff of these designs is the reduced realism and resulting uncertainty around whether identical results would hold for participants' real emotional situations. The present design addresses these concerns regarding the generalizability of prior findings to actual emotional situations, as well as concerns regarding whether participants truly desired empathy when immersing themselves in imaginary scenarios. We pre-registered the following hypotheses:

H1: Participants would choose to receive empathy responses from humans more than AI.

H2: Participants would rate AI empathy responses as more empathetic and making them feel more heard than human responses.

The present study was conducted in two parts. In part 1, we recruited an initial sample of 151 participants from Prolific who were screened for English fluency using an inbuilt Prolific pre-screener. The samples size was calculated using G*Power 3.1 to detect a small-medium effect for a paired samples and one sample t-test with 95% power, plus additional over-sampling to account for 25% participant dropout between part 1 and part 2 based on similar prior work[12]. We excluded three participants total, two for using AI (identified by copy and pasted prompt fragments across their responses), and an additional participant for responding only in sentence fragments that were not personal scenarios. As pre-registered, we additionally excluded two individual trials from the part 1 choice analyses in which the responders did not share a scenario. This left a final sample of 148 participants (95 female, 52 male, 1 nonbinary; $M_{age} = 38.39$ years, $SD_{age} = 13.13$; 75 White/Caucasian, 64 Black/African American, 2 Hispanic/Latino, 3 Asian/Pacific Islander, 1 Native American, 1 other, 2 mixed).

In part 1, we asked participants to share six different situations from their lives "for which [they] would appreciate receiving an empathetic response." Participants were asked to briefly describe the situation as if they were discussing it with someone who would provide an empathetic response. After they wrote about each scenario, they were asked whether they would prefer to receive an empathetic response from a human or AI for each situation. Participants were also told ahead of time that, following this task, we would generate real empathy responses for two of their scenarios, from the sources of their choosing, and return these empathy responses to them. This meant that their choices of empathizer were consequential as they would later receive responses corresponding to their actual choices.

In order to reduce the burden of freely recalling six different scenarios for which they desired empathy, we prompted participants with different domains and valence of experience (e.g., "Think of a positive moment in your work, studies, or creative efforts."). We asked participants to think about both positive and negative situations from work, their relationships, and personal goals/struggles. Each participant shared both a positive and negative experience from each of these three domains, for a total of six scenarios. Specific prompt instructions can be found in the supplement (p. 8). Differences in choice between these domains were not of particular interest to us, although we do report exploratory analyses of these in the supplement (pp. 35–38).

Following part 1, in preparation for part 2, we randomly selected one scenario from each participant for which they chose a human empathizer, and one scenario for which they chose an AI empathizer. We then proceeded to generate human empathy responses for these randomly selected scenarios in which participants chose human, and AI responses for the selected scenarios in which participants chose AI. If they only chose one source throughout all their trials, we only generated a single response corresponding to the source they chose. AI responses were created using the ChatGPT-4o through the OpenAI API[20], and human responses were created by Prolific users. Each Prolific user generated an empathy response for a single part 1 participant, such that all participants from part 1 received an empathy response from a unique human empathizer. Both sources were prompted to "write an empathetic response to this person in which you express warmth, care, and share in their experience." Additional details regarding this response generation procedure can be found in the supplement (p. 11).

Next, in part 2, we re-recruited the participants from part 1 ($n = 140$; ~95% retention) to rate these empathy responses created for their unique scenarios. There were approximately eight hours between the launch of part 1 and the launch of part 2, and we left recruitment for part 2 open for 24 hours. In part 2, we presented participants with the two randomly selected scenarios they had shared previously, followed by the human or AI response we had created for each scenario; though if participants selected a single source throughout all of part 1, as stated above, we only generated a single response, so these participants saw only a single scenario-response pair instead of two. Scenario-response pairings were presented one at a time, and we counterbalanced whether participants viewed the human or AI scenario-response pair first. After reading each scenario and the corresponding response which, as in previous studies, was transparently labeled as having been created by a human Prolific user or AI, participants rated the responses using the same metrics we used previously: "The response was empathetic", "The response was compassionate", "The response was caring", "Reading this response makes me feel comforted", "Reading this response makes me feel cared for", "Reading this response makes me feel validated", "I feel connected to the responder", "The response felt authentic." Participants also answered the same measure regarding responder effort (i.e., "How much effort do you think the responder put into their response?"). We use multilevel modeling, an aggregated one-sample t-test, and paired samples t-test as our primary analytic methods. Exploratory analyses and details about additional measures can be found in the supplemental materials (pp. 5 and 35–38).

## Transparency and Openness

We report our sample size determination, all manipulations, and all data exclusions. Study 1 was not pre-registered as it was primarily exploratory, however we pre-registered the designs, exclusion criteria, and analysis plans for Study 2 (https://osf.io/9ucyr/), Study 3 (https://osf.io/83spu/), and Study 4 (https://osf.io/9jakn/) on 11/27/24, 12/04/24, and 06/16/25 respectively.

Data were analyzed using R version 4.4.2[25], all plots were created using ggplot2 version 3.5.1[26], and all multilevel models were fit using the lmerTest R package version 3.1.3[27]. Multilevel modeling, like most linear modeling, technically assumes independence and normality of residuals within each level, although it is generally robust to violations of this assumption. Across all studies, data distributions were assumed to be normal but this was not formally tested.

Informed consent was obtained for all participants. Participants recruited from Prolific (Studies 1 and 4) were compensated at a rate of $12/hour (see supplemental materials p. 11 for more on Study 4 compensation) and participants recruited from an undergraduate research pool (Studies 2 and 3) received undergraduate course credit for their participation. Samples were distinct across studies. All methods were approved by the Pennsylvania State University Institutional Review Board.

### Reporting summary

Further information on research design is available in the Nature Portfolio Reporting Summary linked to this article.

## Results

All analyses are grounded in a causal inference framework to choose appropriate control variables and obtain estimates of causal effects[28]. The supplemental materials contain a more detailed discussion of causal inference (p. 14), as well as our theorized causal structure for each study depicted as directed acyclic graphs (DAGs; Figs. S1, S4, S8, and S10)[29]. These DAGs identify the set of control variables that must be included in each model for

appropriate inference of causal effects (i.e., the sufficient adjustment set)[30,31]. For simplicity, we do not list each adjustment set in the main text, but all analyses include these variables as required by the DAG. All models are available in the analysis code, and we additionally discuss the adjustment sets used for the main text analyses in the supplemental materials (pp. 20, 26, 33, and 36).

### Study 1

**Choice.** For the initial analysis, we converted each individual's choices across the 20 trials into a proportion of trials in which the human response was chosen. A one-sample t-test comparing the proportion of trials in which human was chosen to an a priori value of 0.5 (indicating chance, or no choice preference), revealed that participants had a significant preference for human responses, selecting the human response an average of 57.1% of the time, $t(151) = 5.37$, $p < 0.001$, $\eta_p^2 = 0.16$, 95% CI [0.55, 0.60] (see Fig. 1). Additionally, we found that 61.18% of participants chose human in the majority of trials (>10 trials out of the 20 total trials), 25% of participants chose AI in the majority of trials, and 13.82% chose them equally (chose human in10 trials and AI in 10 trials). We additionally examined this in the context of a multilevel logistic model nesting trials within participant. A random intercept significantly improved model fit ($p < 0.001$) and ICC = 0.08, indicating the suitability of multilevel modeling. The significance of the intercept in this approach suggested that participants were significantly more likely to choose human over AI responses, $b = 0.31$, $z = 5.30$, $p < 0.001$, 95% CI [0.19, 0.42].

**Response Quality.** We approached this analysis suspecting that the six trial-level rating variables of response quality would have a highly similar pattern across individual rating types. To test this, we performed a one-factor confirmatory factor analysis (CFA). While the CFI was in an acceptable range (0.932) and the SRMR indicated good fit (0.029), the RMSEA suggested inadequate fit (0.227). We consulted the modification indices which suggested allowing the error variances of the ratings "The response was empathetic" and "The response was compassionate" to covary. These items have a similar wording structure, and both refer to judgments of qualities of the responses themselves rather than participants' emotional reactions to the responses (e.g., "Reading this response makes me feel cared for"). Because of this, we found the covariance in errors for these particular items to be justifiable. This updated model had good fit: CFI = 0.995, RMSEA = 0.062, SRMR = 0.007. There was also high reliability ($\omega_u = 0.95$)[32], so we averaged these six items into a single response quality composite and report analyses only on this composite,

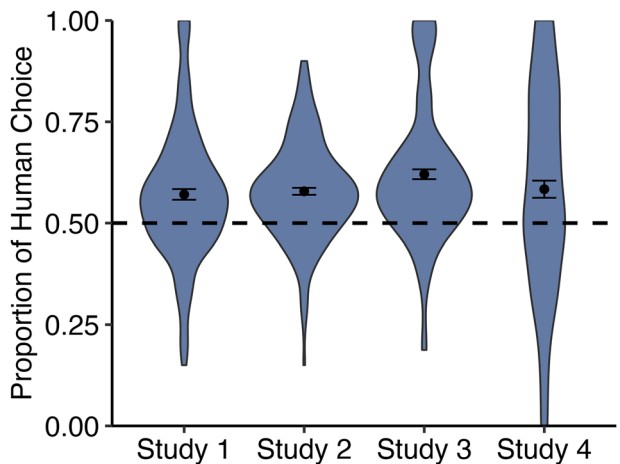

**Fig. 1 | Proportion of human choice across trials for all studies.** Violin plots display the proportion of human choice across all trials for each study. The dashed line represents an equal proportion of choosing human and AI. The sample sizes for each plot are: $n = 152$ for Study 1, $n = 196$ for Study 2, $n = 195$ for Study 3, and $n = 148$ for Study 4. Points indicate the mean, and bars represent standard errors.

**Fig. 2 | Average trial ratings across trials for Study 1.** Violin plots display the spread of participant-level average ratings across trials for Study 1, split by source. The sample size for all AI ratings (white) is $n = 147$, and the sample size for all human ratings (blue) is $n = 152$. Points indicate the mean of participant averages, and bars represent standard errors.

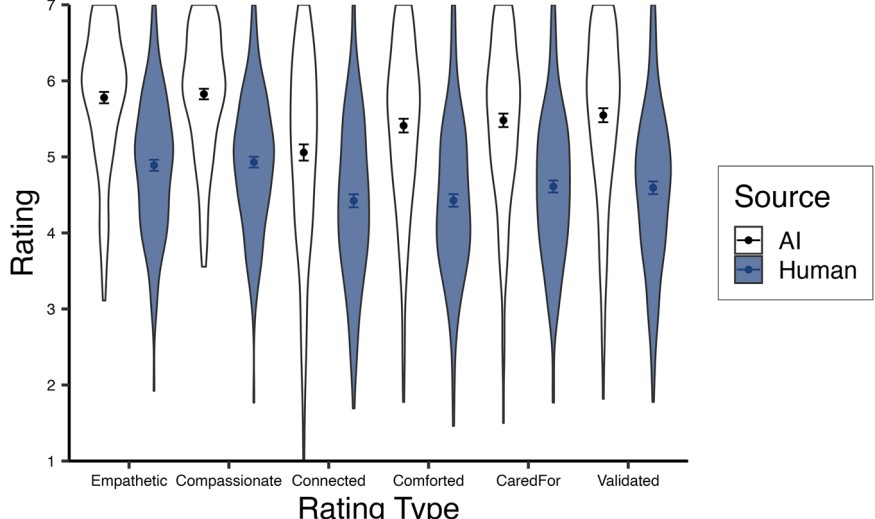

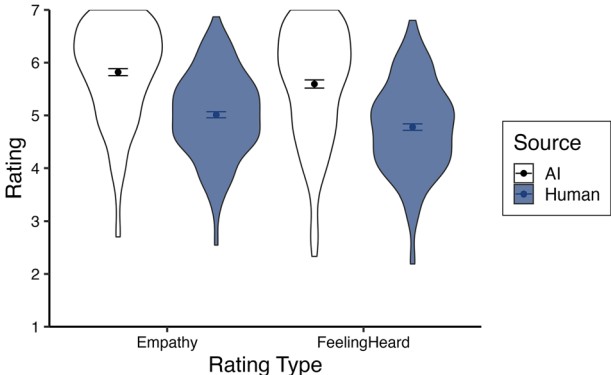

**Fig. 3 | Average empathy and feeling heard ratings across trials for Study 2.** Violin plots display the spread of participant-level average ratings across trials for Study 2, split by source. The sample size for both AI ratings (white) is $n = 196$, and the sample size for both human ratings (blue) is $n = 196$. Points indicate the mean of participant averages, and bars represent standard errors.

but note that the general pattern of results holds for all rating items individually (see Fig. 2 for individual ratings).

As our primary analysis of response quality ratings, we used multilevel modeling. A random intercept significantly improved model fit ($p < .001$) and ICC = 0.22, indicating the suitability of multilevel modeling. To isolate the within-person effect of response source, we used a centered-within-context (i.e., person-mean-centered) contrast coded predictor (AI = -0.5, human = 0.5; for details of centering, see supplement p. 19). We additionally included a random slope for this predictor which significantly improved model fit ($p < 0.001$). This analysis found that AI responses were rated significantly higher on our composite rating than human responses, $b = -0.89$, $t(140.48) = -9.58$, $p < .001$, $\sim R^2 = 0.22$, 95% CI [-1.07, -0.71]. The reported effect size can be interpreted as the proportion of residual within-person variance explained by including response source as a predictor[33].

**Effort.** Finally, we report an additional exploratory analysis examining the relationship between the block-level measure of perceived effort of the responder and response source. These ratings were made separately for human and AI responses at the end of each block. We found that AI responses were perceived as significantly more effortful than human responses, $b = -0.61$, $t(138) = -4.55$, $p < .001$, $\eta_p^2 = 0.13$, 95% CI [-0.87, -0.34]. We also found that the more effortful participants perceived human responses relative to AI, the more they tended to choose human responses, $b = 0.03$, $t(137) = 4.32$, $p = .042$, $\eta_p^2 = 0.12$, 95% CI [0.02, 0.04], and the higher they rated the response quality of human responses relative to AI, $b = 0.43$, $t(137) = 8.53$, $p < .001$, $\eta_p^2 = 0.35$, 95% CI [0.33, 0.53]. Note that these relationships are bidirectional and equally imply that the more effortful participants perceived AI responses relative to human, the more they tended to choose AI responses and rated AI responses as higher quality relative to human.

## Study 2

**Choice.** As in Study 1, we initially analyzed choice by converting each individual's choices into a proportion of trials in which human was chosen and comparing to an a priori value of 0.5. Supporting our first pre-registered hypothesis, we found evidence of a significant preference for human responses, selecting them an average of 57.8% of the time, and replicating the previous study $t(195) = 9.10$, $p < .001$, $\eta_p^2 = 0.30$, 95% CI [0.56, 0.60] (see Fig. 1). Additionally, we found that 69.39% of participants chose human in the majority of trials (>10 trials out of the 20 total trials), 16.32% of participants chose AI in the majority of trials, and 14.29% chose them equally (chose human in 10 trials and AI in 10 trials). We also planned to examine this using a multilevel logistic model nesting trials within participants; however, the ICC was low (0.01) and a random

intercept did not significantly improve model fit ($p = 0.08$), indicating the unsuitability of multilevel modeling. Consequently, proceeding with standard logistic regression, a significant intercept further supported our first pre-registered hypothesis, $b = 0.32$, $z = 9.77$, $p < 0.001$, 95% CI [0.25, 0.38].

**Response Quality.** As stated in our pre-registration, we performed a two-factor CFA regressing our rating measures "The response was empathetic", "The response was compassionate", and "The response was caring," onto a factor representing "Empathy", and rating measures "Reading this response makes me feel comforted", "Reading this response makes me feel cared for", and "Reading this response makes me feel validated," on a second factor representing "Feeling Heard." We decided this based on our appraisals of the constructs and the error covariance in Study 1 between "The response was empathetic" and "The response was compassionate." Similar to our justification of this error covariance in Study 1 suggested by the modification indices, we reasoned that the items related to our "Empathy" construct represent a judgment of the qualities of the response itself (e.g., describing *the response* as empathetic), while the other items corresponding to "Feeling Heard" referred to the response recipient's internal state (e.g., *the recipient* feels comforted). Compared to our overall aggregation in Study 1, we removed "I feel connected to the responder" from these factors as we believed it could represent a construct separate from "Feeling Heard," (and was similarly treated as distinct in prior work[12]) and added an additional item for the "Empathy" factor, "The response was caring," so that each factor would have three items. This pre-registered factor structure had overall good fit: CFI = 0.994, RMSEA = 0.077, SRMR = 0.007. The reliabilities for each factor were high (Empathy $\omega = 0.96$, Feeling Heard $\omega = 0.95$), so we averaged the corresponding rating items (three for "Empathy," three for "Feeling Heard") into two composite ratings and analyzed only these.

We examined the ratings of empathy and feeling heard using multilevel modeling, again nesting trials within participants. A random intercept significantly improved model fits ($p$'s $<0.001$) and ICC's = 0.20 for each dependent variable. We first analyzed the effect of response source on empathy and feeling heard using a centered-within-context contrast code (AI = -0.5, human = 0.5) with a random slope which significantly improved model fits ($p$'s $<.001$). Replicating Study 1 and in support of our second pre-registered hypothesis, we found that AI responses were rated as more empathetic than human responses, $b = -0.83$, $t(196.86) = -11.25$, $p < .001$, $\sim R^2 = 0.21$ 95% CI [-0.97, -0.68], and made participants feel more heard, $b = -0.84$, $t(195.80) = -9.84$, $p < .001$, $\sim R^2 = 0.22$, 95% CI [-1.00, -0.67] (see Fig. 3).

**Effort.** We then explored the role of perceived effort of the responder both as an outcome and predictor. First, we analyzed influences of higher/lower effort perception by treating effort as an outcome in a multilevel model (random intercept significantly improved model fit, $p < .001$, ICC = 0.18). AI responses were perceived as significantly more effortful than human responses, $b = -0.78$, $t(194.25) = -11.57$, $p < 0.001$, $\sim R^2 = 0.28$, 95% CI [-0.91, -0.65], replicating the block-level effect from Study 1 but now specified on a trial-level basis that takes participants' choices into account (see Fig. 4).

Finally, we analyzed the role of perceived responder effort in making responses appear empathetic, and response recipients feel heard. Using a centered-within-context predictor for effort, we found that higher perceived effort strongly predicted higher ratings of empathy, $b = 0.80$, $t(3439.47) = 52.01$, $p < .001$, $\sim R^2 = 0.50$, 95% CI [0.77, 0.83], and feeling heard, $b = 0.91$, $t(3534.60) = 53.37$, $p < .001$, $\sim R^2 = 0.50$, 95% CI [0.87, 0.94]. We also found a significant interaction between effort and response source for both empathy, $t(3623.08) = 12.33$, $p < .001$, 95% CI [0.35, 0.48], and feeling heard, $t(3670.19) = 9.83$, $p < .001$, 95% CI = [0.30, 0.44]. Effort more strongly predicted ratings for human responses relative to AI (see Fig. 5), although effort was a significant predictor for both human (empathy: $b = 1.00$, $t(3603) = 52.34$, $p < .001$, 95% CI [0.97, 1.04]; feeling heard: $b = 1.09$, $t(3593) = 51.67$, $p < .001$, 95% CI [1.05, 1.13]) and AI sources (empathy: $b = 0.59$, $t(3239) = 22.58$, $p < .001$, 95% CI [0.54, 0.64]; feeling heard: $b = 0.72$, $t(3424) = 24.95$, $p < .001$, 95% CI [0.67, 0.78]).

## Study 3

**Choice.** We compared the proportion of trials in which human was chosen to an a priori value of 0.5 and, supporting our first pre-registered hypothesis, found evidence of a preference for human responses, selecting them an average of 62.1% of the time, $t(194) = 9.94$, $p <0.001$, $\eta_p^2 = 0.34$, 95% CI [0.60, 0.64] (see Fig. 1). Additionally, we found that 72.31% of participants chose human in the majority of trials (>8 trials out of the 16 total trials), 11.28% chose AI in the majority of trials, and 16.41% chose them equally (chose human in eight trials and AI in eight trials). This descriptively stronger preference for human empathy, compared to the first two studies, may be a consequence of the more curated human stimuli which were optimized to be high-quality empathy stimuli in previous work[11]. We also analyzed choice using a multilevel logistic model nesting trials within participant. A random intercept significantly improved model fit ($p <0.001$) and ICC = 0.09. In further support of our first pre-registered hypothesis, this approach also indicated that participants were significantly more likely to choose human over AI responses, $b = 0.53$, $z = 9.39$, $p <0.001$, 95% CI [0.42, 0.65].

**Response Quality.** We pre-registered the same two-factor structure as in Study 2 and performed a CFA, finding overall good fit: CFI = 0.996, RMSEA = 0.058, SRMR = 0.007. The reliabilities for each factor were also high (Empathy $\omega = 0.96$, Feeling Heard $\omega = 0.94$) so, again, we averaged the corresponding rating items into the two composite ratings and analyzed only these.

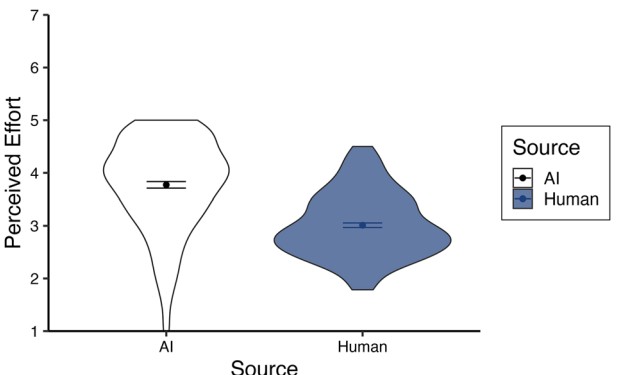

**Fig. 4 | Average effort ratings across trials for Study 2.** Violin plots display the spread of participant-level average ratings across trials for Study 2, split by source. The sample size for AI (white) is $n = 196$, and the sample size for human (blue) is $n = 196$. Points indicate the mean of participant averages, and bars represent standard errors.

Proceeding to the multilevel analysis of empathy and feeling heard, random intercepts were statistically significant for both dependent variables ($p$'s <0.001) and ICC's = 0.33 and 0.32, respectively. Estimating a random slope for source ($p$'s <0.001) and using a centered-within-context contrast code (AI = -0.5, human = 0.5), we found that participants rated AI responses as more empathetic, $b = -0.18$, $t(183.82) = -2.28$, $p = .024$, $\sim R^2 = 0.16$, 95% CI [-0.33, -0.02], and made them feel more heard, $b = -0.21$, $t(182.46) = -2.57$, $p = 0.011$, $\sim R^2 = 0.14$, 95% CI [-0.36, -0.05], though this difference is descriptively smaller than previous studies likely due to the curated human stimuli (see Fig. 6). This supports our second pre-registered hypothesis.

**Effort.** Finally, we explored effects related to perceived effort. First, we analyzed effort as an outcome using multilevel modeling (random intercept $p <0.001$, ICC = 0.27). Replicating our prior studies, AI expressions were perceived as having significantly more effort put into them, $b = -0.32$, $t(180.15) = -4.63$, $p <0.001$, $\sim R^2 = 0.17$, 95% CI [-0.46, -0.19].

Perceived effort significantly predicted both empathy ratings, $b = 0.70$, $t(2781.20) = 44.15$, $p <.001$, $\sim R^2 = .41$, 95% CI [0.67, 0.73], and feeling heard, $b = 0.78$, $t(2744.98) = 46.80$, $p <0.001$, $\sim R^2 = 0.44$, 95% CI [0.75, 0.81]. We also replicated the interaction between response source and perceived effort for both empathy, $t(2927.67) = 4.18$, $p <0.001$, 95% CI [0.08, 0.23], and feeling heard, $t(2922.17) = 4.02$, $p <0.001$, 95% CI [0.08, 0.24]. Replicating Study 2, the relationships of effort and ratings of empathy and feeling heard were significant for both human (empathy: $b = 0.78$, $t(2864) = 35.25$, $p <.001$, 95% CI [0.74, 0.82]; feeling heard: $b = 0.86$, $t(2866) = 36.84$, $p <.001$, 95% CI [0.81, 0.90]) and AI (empathy: $b = 0.62$, $t(2759) = 23.16$, $p <.001$, 95% CI [0.57, 0.68]; feeling heard: $b = 0.70$, $t(2699) = 24.88$, $p <.001$, 95% CI [0.65, 0.76]), but stronger for human than AI expressers.

## Study 4

**Choice.** We compared each participant's proportion of trials in which they chose human to an a priori value of 0.5 and, in support of our first pre-registered hypothesis, again found evidence of a preference for human responders, selecting them an average of 58.4% of the time, $t(147) = 3.95$, $p <0.001$, $\eta_p^2 = 0.10$, 95% CI [0.54, 0.63] (see Fig. 1). Additionally, 47.97% of participants chose human in the majority of trials (>3 trials out of the six total trials), 25.68% chose AI in the majority of trials, and 26.35% chose them equally (chose human in three trials and AI in three trials).

As conducted for the studies above, our pre-registered hypothesis could also be tested by analyzing the intercept in a multilevel logistic model nesting trials within participant. A random intercept significantly improved model fit ($p <0.001$) and ICC = 0.15. In further support of our first pre-registered hypothesis, this multilevel modeling approach similarly suggested that participants were significantly more likely to choose human over AI empathizers, $b = 0.38$, $z = 3.95$, $p <0.001$, 95% CI [0.20, 0.58].

**Fig. 5 | Effort x Source interaction for Study 2.** Study 2 model-predicted relationships estimated from perceived effort ratings one standard deviation above and below the person-mean-centered predictor. The number of observations for AI (gray) is $n = 1609$ for Empathy (left) and n = 1609 for Feeling Heard (right), and the number of observations for human (blue) is $n = 2211$ for Empathy (left) and $n = 2210$ for Feeling Heard (right). Shaded areas represent standard errors.

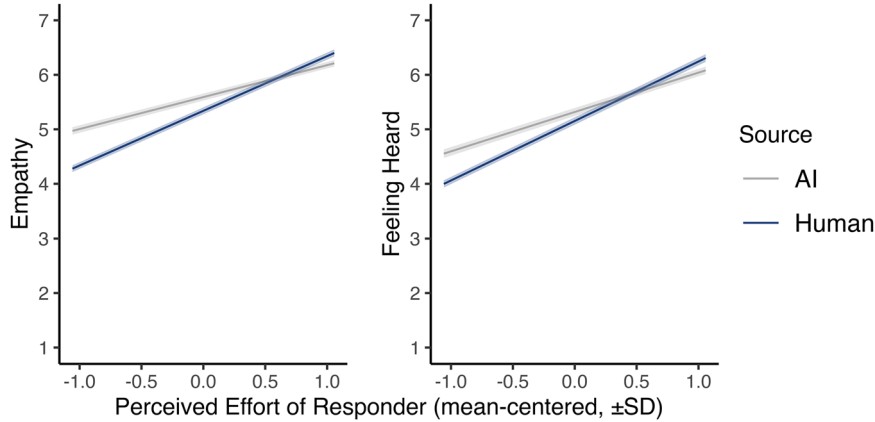

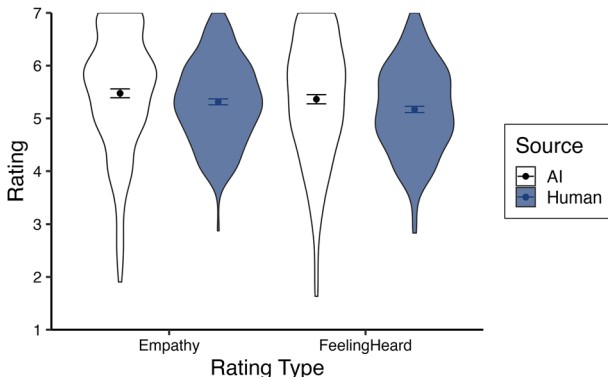

**Fig. 6 | Average empathy and feeling heard ratings across trials for Study 3.** Violin plots display the spread of participant-level average ratings across trials for Study 3, split by source. The sample size for both AI ratings (white) is $n = 182$, and the sample size for both human ratings (blue) is $n = 195$. Points indicate the mean of participant averages, and bars represent standard errors.

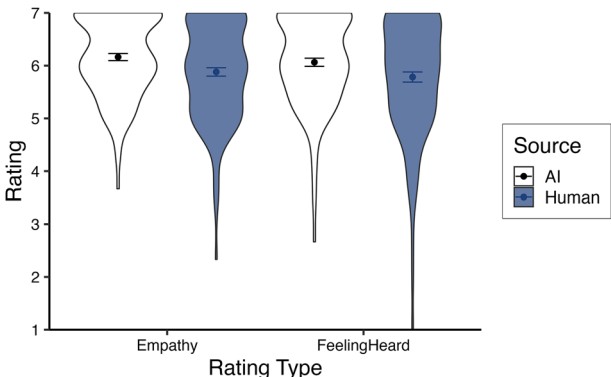

**Fig. 7 | Average empathy and feeling heard ratings across trials for Study 4.** Violin plots display the spread of participant ratings for Study 4, split by source. The sample size for both AI ratings (white) is $n = 140$, and the sample size for both human ratings (blue) is $n = 140$. Points indicate the mean, and bars represent standard errors.

**Response Quality.** As stated in our pre-registration, we performed the same two-factor CFA that we used in Studies 2 and 3. Holistically, we found acceptable model fit (though the RMSEA was slightly high), CFI = 0.989, RMSEA = 0.084, SRMR = 0.017, and the reliabilities for each factor were high (Empathy $\omega = .88$, Feeling Heard $\omega = .91$), so we averaged the corresponding rating items into two composite ratings and analyzed only these.

The test of response source using a centered-within-context contrast code in a multilevel model is fully redundant with a paired samples t-test as there are only two trials per participant. Supporting our second pre-registered hypothesis, a paired samples t-test revealed that participants rated AI responses as more empathetic than human responses, $b = -0.26$, $t(118) = -2.77$, $p = 0.007$, $\eta_p^2 = 0.06$, 95% CI [-0.45, -0.07], and that they made them feel significantly more heard, $b = -0.26$, $t(118) = -2.31$, $p = 0.023$, $\eta_p^2 = 0.04$, 95% CI [-0.47, -0.04] (see Fig. 7).

**Effort.** We also explored the role of perceived effort of the responder as an outcome and predictor, as we have done in the previous studies. Replicating the previous studies, a paired samples t-test revealed that AI responses were perceived as significantly more effortful than human responses, $b = -0.23$, $t(118) = -2.51$, $p = 0.013$, $\eta_p^2 = 0.05$, 95% CI [-0.41, -0.05].

Finally, we analyzed the role of perceived responder effort as a predictor for ratings of empathy and feeling heard. In line with our prior findings, we found that higher effort ratings strongly predicted higher ratings of empathy, $b = 0.72$, $t(114.79) = 10.25$, $p < .001$, $\sim R^2 = 0.47$, 95% CI [0.58, 0.86] and feeling heard, $b = 0.89$, $t(112.48) = 11.43$, $p < .001$, $\sim R^2 = 0.53$, 95% CI [0.74, 1.04]. We did not technically replicate the interaction from Studies 2 and 3 between response source and perceived effort for empathy, $t(126.23) = 1.86$, $p = .066$, 95% CI [-0.03, 0.96], or feeling heard, $t(125.06) = 1.41$, $p = 0.162$, 95% CI [-0.17, 1.02]. Though, in support of the prior studies, follow-up tests revealed an identical pattern as both Studies 2 and 3 such that the relationships between effort and ratings of empathy and feeling heard were significant for both human (empathy: $b = 0.95$, $t(205) = 6.62$, $p < 0.001$, 95% CI [0.67, 1.24]; feeling heard: $b = 1.11$, $t(196) = 6.47$, $p < 0.001$, 95% CI [0.77, 1.44]) and AI (empathy: $b = 0.49$, $t(205) = 3.38$, $p < 0.001$, 95% CI [0.20, 0.77]; feeling heard: $b = 0.68$, $t(196) = 3.97$, $p < 0.001$, 95% CI [0.34, 1.02]), but descriptively stronger for human than AI expressers. The lack of interaction significance is likely a result of the overall lower power of Study 4 compared to the previous studies due to fewer observations.

## General Discussion

We explored the complex nature of empathic AI. There is much discussion of AI empathy and its implications for human society, with some work showing that AI empathy expressions are rated as higher on empathy and quality[1,11,12] despite algorithm aversion[34,35] and ethical worries about human-AI interaction[36,37].

We find support for some descriptive claims made by AI empathy detractors: People prefer receiving empathy from a human over AI when given the option (although there is variability here, as can be seen in Fig. 1, with as many as 25% of our participants from Studies 1 and 4 choosing AI responses most of the time)[6,8]. However, we also find support for AI empathy advocates[9,10]: Consistent with prior work[1,11,12], people rated caring responses from AI as both more empathetic than human responses, and as making them feel more heard. We name this perplexing pattern of results in which people choose human empathy, but rate AI empathy as higher quality, the "AI empathy choice paradox." We replicated the paradox across studies, using different samples and stimuli sets.

We also find evidence to support the descriptive anti-AI empathy claim that part of the value of empathy may be tied to the fact that it is effortful for humans, which serves as a signal of the recipient's importance to the empathizer[6]. Empathy recipients *do* pay attention to effort put into responses which, in turn, impacts how they feel. However, while we find support for the importance of perceived effort, the assumption that human empathy is perceived as more effortful is not supported. We found that AI empathy was perceived as entailing more effort. It is possible that participants view the question of AI effort as a category error, assuming the question is asking participants to answer as if the AI were human. Alternatively, people may authentically perceive effort in the AI responses and participants may not naturally consider the underlying reality that empathy is effortless for AI.

Because of this increased importance of effort for human empathy but generally lower perception of effort relative to AI, effort may also serve as a double-edged sword for human empathy: It really does contribute to making empathy recipients feel cared for, however, average human empathy is not seen as particularly effortful. Taken together, this may highlight the potential value of AI empathy in contexts in which there are limited resources for effortful human empathizing. While the highest echelon of human empathy may have an advantage over AI, this requires tremendous effort which may not always be possible for most empathizers, most of the time. Still, this is speculative and future work needs to be done to explore what truly underlays this judgment of AI as effortful.

We also note that our finding of AI empathy avoidance may not be specific to empathy per se, but rather a symptom of a larger algorithmic aversion[34,35]. Previous work has found an aversion to receiving advice from AI regarding societal issues[38], a variety of personal topics including dating/relationships[39], and moral dilemmas[40]. Nonetheless, we do not think that our findings must reflect a domain-specific empathy avoidance effect (for similar claim about empathic vs. non-empathic effort, see here[19]). Our finding about human empathy preference may reflect generalized algorithm

aversion[41], and our results can be practically impactful, even if they are reflective of a broader AI aversion, as they unpack the application of this aversion in an empathic context.

## Limitations

While we find a consistent general pattern of viewing AI empathy more positively than human empathy (in terms of empathic quality and the recipient's emotional reactions to the empathy), it should be acknowledged that there are potential limitations. For one, although we provide evidence of the choice paradox for situations related to participants' own lives, much of the information around the empathizer is decontextualized. AI empathy may make people feel more heard than empathy from a human stranger, but how does it compare to empathy from a human with whom the recipient has a relationship?

However, this question could also be reversed in that one could ask how human empathy compares to empathy from an AI with whom the recipient has a relationship. While not necessarily common, there are social AI chatbots designed specifically for relationship contexts. One such example, Replika, is designed for users to form friendships, or even romantic relationships[42]. Perhaps a less extreme variant of a similar question could be: How does the value of AI empathy change when exposed to it over time, in the context of repeated interactions[43]? The avoidance of AI empathy may decrease, as there is some evidence to suggest that previous positive exposure to AI can reduce aversion to it[38].

It is also worth exploring variations in context regarding the nature of interaction itself. For example, how does empathy value change (for human or AI) if the responses are spoken by the human empathizer or one of the ChatGPT voice options? Or if the empathizer is embodied in some way (e.g., in the same room)? New possibilities for embedding real-time chatbot dialogs within survey paradigms may also afford immersive engagement[44] and attempts to elicit empathetic reactions in real time. Further, as AI technology develops and changes, both in form (i.e., voice versus text) as well as simple model updating, it leads to questions regarding how its empathic abilities will differ in the future. It may be important to understand how these elements of the empathic context are weighed against one another, as often our choice of empathizer may be asymmetrical (e.g., an embodied human versus AI voice).

More work also needs to be done exploring features of the empathy recipient and particular situation that may impact participants' choice of empathy source, and the value derived from each source. Future work should seek to explore individual characteristics that may impact the desire to seek out AI empathy (previous AI use, loneliness, tendency to anthropomorphize). We also encourage future work to test features of the empathy-evoking situations which may impact these choices and subsequent perceptions of empathic quality. We begin to explore these in the supplement where we find, for example, that the empathic quality advantage of AI was greater for emotional (vs. physical) suffering (pp. 20, 26–27), and for compassion (vs. empathy; pp. 21, 27–28), but further work stands to be done here.

A final limitation of our studies could be our choice of trial-level questions. It is possible that we do not capture the full range of emotional experiences when receiving an AI empathy response. Most of our questions ask participants to make positive judgments regarding the responses, but if we also asked about negative evaluations of AI empathy that might capture unfamiliarity or uncanniness (e.g., "disturbing", "distressing", "frightening"), it may also score high on those measures, indicating some level of ambivalence. We found some evidence of certain domains in which, at the very least, AI does not robustly outperform high-quality human empathy: connectedness to the empathizer, and perceived authenticity of the response (discussed in the supplement, pp. 29, 33, and 37).

## Conclusion

Our research serves as a major step towards resolving a fundamental question in human-AI interaction with empirical evidence rather than speculation: People prefer human empathizers, yet consistently rate AI responses as more empathetic and report feeling more heard when they do engage with them.

This AI empathy choice paradox—preferring one source while benefiting more from another—illuminates complex dynamics in how we seek and receive emotional support. This paradox challenges both AI enthusiasts and skeptics: while individuals initially avoid AI empathy, they benefit from it when experienced, suggesting the potential value of AI empathy, particularly in contexts where consistent, high-quality human empathy is unavailable.

As AI becomes increasingly integrated into emotional support domains, society faces the challenge of harnessing its capacity for high-quality empathic expression while honoring human agency in choosing support sources. Rather than viewing AI empathy as replacing human connection, our research points toward a future where AI augments human empathy—filling gaps when human support falls short while preserving the option for human connection that many people prefer.

## Data availability

All data, stimuli, and supplemental materials are available at the OSF repository: https://osf.io/3q7ke.

## Code availability

All analysis code is available at the OSF repository: https://osf.io/3q7ke.

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

## Acknowledgements

This material is based upon work supported by the National Science Foundation Graduate Research Fellowship Program under Grant No. DGE1255832. Any opinions, findings, and conclusions or recommendations expressed in this material are those of the author(s) and do not necessarily reflect the views of the National Science Foundation. This work was additionally funded by The Pennsylvania State University Rock Ethics Institute, and Consortium on Moral Decision-Making. We acknowledge support from the Natural Sciences and Engineering Research Council of Canada RGPIN-2019-05280. The funders had no role in study design, data collection and analysis, decision to publish or preparation of the manuscript. A special thanks to the following undergraduate research assistants for their support in developing human empathy stimuli: Rachel Buterbaugh, Jessica Firestone, Greg McLauchlan, Skylar Cui, and Evan Butler. An additional thanks to Ovsyannikova and team[11] for allowing us to adapt their stimulus set, and to Yidan Yin for her advice on Study 4 implementation. The present work was based on the lead author's master's thesis. The work has been presented at the Moral Psychology Research Group Meeting, at the University of Virginia, the Society for Social and Personality Psychology Annual Convention, the Society for Affective Science Annual Conference, and the Annual Meeting of the Academy of Management (and the abstract published in the Academy of Management Proceedings).

## Author contributions

J.D.W. and C.D.C. were responsible for conceptualization, methodology, data curation, analysis, funding acquisition, and writing (original draft, review and editing). C.D.C. was additionally responsible for supervision. M.I. was responsible for conceptualization, methodology, supervision, and writing (review and editing).

## Competing interests

The authors declare no competing interests.
