## [Transparent Peer Review file · Communications Psychology]

People choose to receive human empathy despite rating AI empathy higher

Corresponding Author: Mr Joshua Wenger

Version 0:

Decision Letter:

Dear Mr Wenger,

Thank you for your patience during the peer-review process. Your manuscript titled "The AI Empathy Choice Paradox: People Prefer Human Empathy Despite Rating AI Empathy Higher" has now been seen by 2 reviewers, and I include their comments at the end of this message. They find your work of interest but raised some important points. We are interested in the possibility of publishing your study in Communications Psychology, but would like to consider your responses to these concerns and assess a revised manuscript before we make a final decision on publication.

We therefore invite you to revise and resubmit your manuscript, along with a point-by-point response to the reviewers. Please highlight all changes in the manuscript text file.

Editorially, we suggest that you provide further clarification regarding the concepts and methodology. In particular, please specify how you define "empathy" and how you measure the "empathic response". Please provide additional detail on the pilot study and pre-testing of the human and AI responses for equivalence.

I am attaching an Editorial Requests Table that details critical reporting requirements for the revised manuscript. Please attend to each item and ensure your manuscript is fully compliant. If your revised manuscript is not aligned with these requests on major issues, such as those concerning statistics, it may be returned to you for further revisions without re-review.

Please submit the following items:

- Revised manuscript
- Point-by-point response to the referees' comments
- Cover letter (as a separate document)
- <https://www.nature.com/documents/nr-reporting-summary.pdf>>Nature Research Reporting Summary
- Completed Editorial Request Table (attached).

via this link: Link Redacted .

Additional guidance is available in our style and formatting guide Communications Psychology formatting guide.

Best regards,

Yafeng Pan

Yafeng Pan, PhD
Editorial Board Member
Communications Psychology
orcid.org/0000-0002-5633-8313

REVIEWER EXPERTISE:

Reviewer #1: AI, Empathy

Reviewer #3: AI, Empathy

REVIEWER REPORTS:

Reviewer #1 (Remarks to the Author):

I would like to thank the editor and the authors for the opportunity to review this timely and well-crafted manuscript. The paper is thoughtful, clearly written, and sensitively presented, with particular strength in its attention to participants' preferences, which enhances the ecological validity of a hotly debated and important topic. The authors establish an important and replicable phenomenon, framed as the "AI empathy choice paradox," which is both novel and likely to be of significant interest to the field and advances our understanding of how people perceive AI-generated empathy compared with human responses. The introduction is clearly written, and the research goals are well defined. Though some aspects, such as the operationalisation of empathy and the interpretation of results, may benefit from further clarification, the results, supported by pre-registered hypotheses, are analysed and reported with rigour, and I appreciate the code sharing and the application of the DAG framework. The conclusions portrayed are generally convincing and the discussion is well-integrated with the broader literature. While I am highly enthusiastic about this work, I do have several major concerns that I believe can be addressed in a revised version. I hope the following comments will assist the authors in strengthening the manuscript.

1. While I agree with the importance of highlighting the recipient's perspective, I do not believe that opponents of the authors' view are dismissive of the idea that "what may matter more is how a recipient feels..." (p.3). For example, although the authors cite Perry (2023), more recent work from Perry's group (Rubin et al., 2025), which is also covered in the paper, explicitly considers the recipient perspective and shows how important it is to consider whether empathy is "fake" or not within this context. Moreover, while the work in this manuscript provides some support for the authors' position in favour of AI-supported empathy, it also provides evidence to the contrary (e.g., showing that human empathy was preferred). Thus, while the authors claim that "there is a reasonable argument to be made in favor of sidestepping the question regarding AI empathy's 'real-ness'" (p.3), there is also a reasonable and valid argument not to do so. As such, I believe that in the Introduction section, both approaches should be presented more evenly. Additionally, each perspective leads to different hypotheses regarding the current studies, and making this explicit would help readers better appreciate the contribution of the work.

2. The definition of empathy should be elaborated further. The authors state that "empathy includes multiple components, such as sharing others' experiences, perspective-taking, and empathic concern" (line 40), but these aspects are not clearly defined. This clarification is methodologically important, as the distinction between components is central to the way empathic responses were later provided to participants. In particular, the authors differentiate between compassionate and empathetic responses. However, some may argue that compassion is indistinguishable from empathic concern, thus requires some clarification.

3. More critically, the instruction for the empathetic response in Study 1-2 focuses on sharing the experience with others.

However, this seems somewhat distinct from common definitions of empathy, in which what is usually considered “shared” is the emotional experience, not the experiences in general (e.g., see the different definitions reviewed by Cuff et al., 2016). This nuance has important implications, as it substantially alters the meaning of the empathic response. To illustrate this, I tested one of Burneau’s vignettes using ChatGPT with the same instructions given to participants in Study 1. When prompted for “an empathetic response in which you share in the experience of the person,” the output differed markedly from the response generated when asked for “an empathetic response in which you share the emotional experience of the person.” As expected, the latter focused more on shared emotions and less on the specific situation, which I believe captures the essence of empathy more accurately (I attach the results from ChatGPT below as a reference):

Empathetic response (sharing the experience)

"That must feel amazing—to finally wake up without that familiar back pain after so long. Not needing Advil first thing in the morning must make the whole day feel lighter and easier."

Empathetic response (sharing the emotional experience):

"I can imagine the sense of relief and even joy you must have felt waking up pain-free. After carrying that discomfort day after day, having a morning like this must feel almost freeing, like you can finally breathe a little easier."

This issue does not necessarily undermine the main conclusions of the paper (since it was typically used as a combined “empathy” factor), but it does raise questions about the validity of what was actually manipulated, and I believe should be acknowledged.

4. I was wondering about the long-term replicability of the study, given that ChatGPT is continually being updated and modified. It may be worth briefly discussing this issue in the limitations section. It could be helpful to provide a few example responses from both human and AI-generated outputs (as a supplement) so that readers can transparently assess these responses. I understand there may be ethical constraints, particularly in Study 4, but this seems especially relevant for Studies 1 and 3.

5. I was also wondering about the training (if any) provided to the responders (and perhaps ChatGPT). Were they given any guidance on the distinction between empathy and compassion beyond the instructions provided in the text? While this distinction may be clear for professionals, non-trained individuals might conflate the two. The same concern applies to participants, who may have their own interpretations of compassion and empathy when providing ratings.

6. I apologise if I have overlooked it, but I was unable to locate the full vignette lists used in Study 1, which the authors mention are in the Supplementary Information.

7. I understand the Study 1 was not pre-registered, but could the authors clarify their hypotheses for this study? If it was entirely exploratory, the authors should make it explicit.

8. Please provide a description of the statistical analyses in the Method sections for each study.

9. I was very much surprised by the finding that participants perceived AI as consistently more effortful. This finding almost seems like it should have served as a sanity check, since why would participants think AI would exert more effort than humans? I have read the authors’ discussion of this in the limitations section, but I remain unconvinced that it is not, as they also suggested, “a category error” resulting from how participants interpret the question. The authors’ alternative explanation for this is intriguing, and may point to something broader about the nature of human-AI relationship, which connects to their later discussion of these interactions. However, since the current study does not allow a definitive conclusion between the two proposed explanations, I recommend discussing this more cautiously and framing it as a question that requires further investigation in future research.

10. Please refer to the supplementary analyses more clearly within the main text (e.g., by specifying the relevant pages and table/figure numbers). This would make the Supplementary Information easier to assess, as it is currently somewhat difficult to follow.

Reviewer #3 (Remarks to the Author):

Thank you for this opportunity to review the manuscript, “The AI Empathy Choice Paradox: People Prefer Human Empathy Despite Rating AI Empathy Higher”. The topic is timely, experimental design is carefully crafted, and the analytic approach is appropriate. I enjoyed reading the manuscript and appreciate the opportunity to be one of the first readers. I believe this study paves the way for developing insights into the role of empathy in human–AI interactions from a psychological perspective. I do not think a major revision is required; however, I would like to offer a few comments.

1. As stated by the authors (p. 6), the study builds on motivated empathy perspectives by adapting the empathy selection task. The motivated empathy perspectives are about the extent to which people feel empathy for others can be malleable and socially motivated. On the other hand, this study examines how much empathy people want to receive. That is, it takes the recipient-focused perspective rather than expresser- (empathizer)-focused perspective. It would be helpful to further elaborate on how the present findings relate to or extend the motivated empathy framework. For instance, could the

motivation to receive empathy be shaped by similar social and contextual factors that drive the motivation to feel empathy toward others?

2. In the pilot study for Studies 1, 2, and 4 (generation of human and AI empathic responses), were the human and AI responses compared in terms of empathy, quality, and emotional reactions to confirm their equivalence? Ideally, individual raters should have been randomly presented with both human- and AI-generated responses, without any labels indicating their source, and asked to rate them. Otherwise, it could be that the AI-generated responses were actually superior to the human-generated ones at baseline.

3. In real-world applications of generative AI, the AI agent is often anthropomorphized. For example, it is typically given a name (e.g., Alexia) rather than being labeled simply as "AI." In contrast, the use of explicit labels ("human" vs. "AI") in the present study may have accentuated the perceived differences between the two, reinforcing categorical distinctions and potentially fostering negative attitudes toward AI among those who hold less favorable attitudes toward algorithmic systems.

Communications Psychology is committed to improving transparency in authorship. As part of our efforts in this direction, we are now requesting that all authors identified as 'corresponding author' create and link their Open Researcher and Contributor Identifier (ORCID) with their account on the Manuscript Tracking System prior to acceptance. ORCID helps the scientific community achieve unambiguous attribution of all scholarly contributions. You can create and link your ORCID from the home page of the Manuscript Tracking System by clicking on 'Modify my Springer Nature account' and following the instructions in the link below. Please also inform all co-authors that they can add their ORCID to their accounts and that they must do so prior to acceptance.

Version 1:

Decision Letter:

Dear Mr Wenger,

Your manuscript titled "The AI Empathy Choice Paradox: People Prefer Human Empathy Despite Rating AI Empathy Higher" has now been seen by our reviewers, whose comments appear below. In light of their advice I am delighted to say that we are happy, in principle, to publish a suitably revised version in Communications Psychology.

We therefore invite you to revise your paper one last time to address the remaining concerns of our reviewers and a list of editorial requests. At the same time we ask that you edit your manuscript to comply with our format requirements and to maximise the accessibility and therefore the impact of your work.

EDITORIAL REQUESTS:

SUBMISSION INFORMATION:

OPEN ACCESS:

Communications Psychology is a fully open access journal. Articles are made freely accessible on publication. For further information about article processing charges, open access funding, and advice and support from Nature Research, please visit <https://www.nature.com/commpsychol/open-access>

* DATA AVAILABILITY:

All Communications Psychology manuscripts must include a section titled "Data Availability" at the end of the Methods section. More information on this policy, is available in the Editorial Requests Table and at <http://www.nature.com/authors/policies/data/data-availability-statements-data-citations.pdf>

Link Redacted

Best regards,

Jennifer Bellingtier

Jennifer Bellingtier, PhD
Senior Editor
Communications Psychology

Yafeng Pan, PhD
Editorial Board Member
Communications Psychology
orcid.org/0000-0002-5633-8313

REVIEWER EXPERTISE:

Reviewer #1: AI, Empathy
Reviewer #3: AI, Empathy

REVIEWERS' COMMENTS:

Reviewer #1 (Remarks to the Author):

Dear Editor,

I would like to thank you and the authors for the opportunity to review this timely and exceptionally well-crafted manuscript. The authors have thoroughly addressed all my concerns, including clarifying the conceptualisation of empathy, refining the description of methods and analyses, expanding the discussion to acknowledge alternative interpretations and limitations,

and providing the generated empathic reactions for transparency. I believe these revisions have further strengthened the manuscript, and in my view, it is now ready for publication in its current form.

Reviewer #3 (Remarks to the Author):

All the points have been thoroughly addressed. Thank you for your careful revisions.

Dear Reviewers,

We would like to express our appreciation for your thoughtful feedback regarding our manuscript, “The AI Empathy Choice Paradox: People Prefer Human Empathy Despite Rating AI Empathy Higher” (COMMSPSYCHOL-25-0749-T), under consideration at *Communications Psychology*. We are grateful for your investment of time and effort in reviewing this manuscript. We found your suggestions insightful, and have served to greatly improve the manuscript. We have implemented changes throughout the paper and supplemental materials, in response to your suggestions. Below, we give point-by-point responses to how we have addressed each concern.

Reviewer #1:

1. While I agree with the importance of highlighting the recipient’s perspective, I do not believe that opponents of the authors’ view are dismissive of the idea that “what may matter more is how a recipient feels...” (p.3). For example, although the authors cite Perry (2023), more recent work from Perry’s group (Rubin et al., 2025), which is also covered in the paper, explicitly considers the recipient perspective and shows how important it is to consider whether empathy is “fake” or not within this context. Moreover, while the work in this manuscript provides some support for the authors’ position in favour of AI-supported empathy, it also provides evidence to the contrary (e.g., showing that human empathy was preferred). Thus, while the authors claim that “there is a reasonable argument to be made in favor of sidestepping the question regarding AI empathy’s ‘real-ness’” (p.3), there is also a reasonable and valid argument not to do so. As such, I believe that in the Introduction section, both approaches should

be presented more evenly. Additionally, each perspective leads to different hypotheses regarding the current studies, and making this explicit would help readers better appreciate the contribution of the work.

Thank you for this suggestion. We have edited these sections of the introduction to cut back on the recipient-focused perspective and expand on the arguments regarding the “fake” nature of AI empathy more specifically:

“However, because AI lacks emotional experience, it is incapable of sharing in the experience of the empathy recipient and has no genuine motivation to alleviate their suffering (Perry, 2023). Therefore, although AI expressions of empathy may seem authentic, they are, in reality, fake and, some have argued, deceptive (Shteynberg et al., 2024). This lack of authentic empathic capacity has led many to be skeptical of its use in the empathy domain (Bloom, 2024; Perry, 2023; Shteynberg et al., 2024).” (pp. 3-4)

Additionally, while we acknowledge that recent work from Perry’s group certainly considers the recipient’s perspective as well, the 2023 paper is still highly influential and lays out a strong argument that notably deemphasizes this (similar to how Inzlicht et al., 2024 deemphasizes the focus on “real-ness”). We do not intend to pitch these as completely opposing perspectives in which one either considers the recipient or does not, but rather competing perspective that are a matter of emphasis. We have edited the language in the concluding sentence in this section to again acknowledge the lack of genuine capacity and better convey this idea of “emphasis” and “focus”:

“Therefore, while acknowledging AI’s lack of genuine empathic capacity, there is also a reasonable argument for deemphasizing the question of “real-ness” and focusing on the question: Does the individual receiving expressions of empathy from AI feel cared for, comforted, validated, etc.?” (p. 4)

2. The definition of empathy should be elaborated further. The authors state that “empathy includes multiple components, such as sharing others’ experiences, perspective-taking, and empathic concern” (line 40), but these aspects are not clearly defined. This clarification is methodologically important, as the distinction between components is central to the way empathic responses were later provided to participants. In particular, the authors differentiate between compassionate and empathetic responses. However, some may argue that compassion is indistinguishable from empathic concern, thus requires some clarification.

Thank you for raising this as we agree that further clarification as to how we differentiate these relative to the existing literature is completely warranted. We have edited the initial definition of empathy in the introduction. Additionally, we have specified more clearly how we are differentiating empathy and compassion in the supplemental materials where we explore this contrast in-depth.

Introduction:

“Empathy includes multiple components, such as experience-sharing, which refers to sharing in the experiences and feelings of the empathy recipient;

perspective-taking, the act of mentally representing and understanding what the recipient is thinking or feeling; and empathic concern or compassion, the emotional response to the recipient's experience and the prosocial desire to reduce their suffering (Zaki & Ochsner, 2012)." (p. 3)

Supplemental Materials:

"As can be seen in our response instructions, we differentiate empathy from compassion by its emphasis on experience-sharing, although it should be noted that this is technically a single facet of the overall construct of empathy (of which compassion could also be considered a facet). Therefore, in the context of discussing the "empathy block" and "empathy response type" we use "empathy" as shorthand for experience sharing." (p. 28, supplement)

3. More critically, the instruction for the empathetic response in Study 1-2 focuses on sharing the experience with others. However, this seems somewhat distinct from common definitions of empathy, in which what is usually considered "shared" is the emotional experience, not the experiences in general (e.g., see the different definitions reviewed by Cuff et al., 2016). This nuance has important implications, as it substantially alters the meaning of the empathic response. To illustrate this, I tested one of Burneau's vignettes using ChatGPT with the same instructions given to participants in Study 1. When prompted for "an empathetic response in which you share in the experience of the person," the output differed markedly from the response generated when asked for "an empathetic response in which you share the emotional experience of the person." As

expected, the latter focused more on shared emotions and less on the specific situation, which I believe captures the essence of empathy more accurately (I attach the results from ChatGPT below as a reference):

Empathetic response (sharing the experience)

"That must feel amazing—to finally wake up without that familiar back pain after so long. Not needing Advil first thing in the morning must make the whole day feel lighter and easier."

Empathetic response (sharing the emotional experience):

"I can imagine the sense of relief and even joy you must have felt waking up pain-free. After carrying that discomfort day after day, having a morning like this must feel almost freeing, like you can finally breathe a little easier."

This issue does not necessarily undermine the main conclusions of the paper (since it was typically used as a combined “empathy” factor), but it does raise questions about the validity of what was actually manipulated, and I believe should be acknowledged.

This is an excellent point and we thank you for raising it. We now acknowledge this explicitly in the section of the supplemental materials where we first explore differences between empathy and compassion. We do also highlight that the difference between the literal experience and the emotional experience may become less clear in some situations within the broader study of empathy, such as empathy for physical pain (which is the focus of the source material for our stimuli). Here, experience-sharing in the pain that

another is feeling is perhaps somewhere between emotional experience-sharing and perspective-taking and includes referents beyond emotionality.

Additional discussion in the Supplemental Materials:

“We additionally mention that our instructions for the empathy responses are phrased as “an empathetic response in which you share in the experience of the person,” though in many definitions of empathy it is the emotional experience, specifically, that is being shared. This distinction between sharing in the general experience versus the emotional experience may be murkier in some cases within the broader study of empathy. For example, a situation in which a person imagines themselves feeling the physical pain and suffering of another may blur the lines between the emotion versus literal experience (as we have in our stimuli; Bruneau et al, 2013). Further, while the identification of the response as an “*empathetic* response” in the instructions may nevertheless imply emotional experience sharing, AI can be sensitive to minor prompting differences, so we raise this as a possible additional consideration in the interpretation of this manipulation.” (pp. 28-29, supplement)

4. I was wondering about the long-term replicability of the study, given that ChatGPT is continually being updated and modified. It may be worth briefly discussing this issue in the limitations section. It could be helpful to provide a few example responses from both human and AI-generated outputs (as a supplement) so that readers can transparently

assess these responses. I understand there may be ethical constraints, particularly in Study 4, but this seems especially relevant for Studies 1 and 3.

Thank you for raising this. We have added a statement acknowledging the ever-changing models and technology:

“Further, as AI technology develops and changes, both in form (i.e., voice versus text) as well as simple model updating, it leads to questions regarding how its empathic abilities will differ in the future.” (p. 35)

Additionally, thank you for pointing out the missing example responses. We have all vignettes and responses for Studies 1-3 in the online OSF repository (<https://osf.io/3q7ke/>) not the supplemental materials document as suggested in the main text. We have clarified this wording.

5. I was also wondering about the training (if any) provided to the responders (and perhaps ChatGPT). Were they given any guidance on the distinction between empathy and compassion beyond the instructions provided in the text? While this distinction may be clear for professionals, non-trained individuals might conflate the two. The same concern applies to participants, who may have their own interpretations of compassion and empathy when providing ratings.

This is an excellent point. The responders and ChatGPT were not given specific instructions beyond the distinction between empathy and compassion made in the prompts. The responses were all screened by the lead author to confirm that it was clear

to this author who knew the specific prompts, whether a response was empathy or compassion. We completely agree that non-trained professionals may not fully be aware of the distinction, and we do see some overlap between the two in the responses as, naturalistically, these tend to co-occur (Depow et al., 2021). This is a useful point to acknowledge, so we have also integrated it into our discussion of these in the supplemental materials, encouraging caution in assuming a perfectly clear divide between empathy and compassion in these responses:

“Additionally, we point out that the separate facets of empathy tend to commonly co-occur in everyday experiences (Depow et al., 2021). We suggest these empathy and compassion manipulations could perhaps be taken at face value rather than as a strict differentiation between empathy and compassion, which may not be completely dissociable in the perception of empathy recipients. Instead, this can be viewed as a manipulation in which an empathizer is prompted to focus on being compassionate (i.e., generating warm feelings of concern) versus empathetic (i.e., experience-sharing) though elements of each may be present in the empathizer’s responding.” (p. 28, supplement)

6. I apologise if I have overlooked it, but I was unable to locate the full vignette lists used in Study 1, which the authors mention are in the Supplementary Information.

Thank you again for pointing this out. See response to point 4; these are in the OSF repository which we have now clarified in the main text.

7. I understand the Study 1 was not pre-registered, but could the authors clarify their hypotheses for this study? If it was entirely exploratory, the authors should make it explicit.

Thank you for catching this, Study 1 was exploratory and we have now identified it as such.

8. Please provide a description of the statistical analyses in the Method sections for each study.

We have added this to each method section, e.g., :

“As described below, we use multilevel modeling and an aggregated t-test as our primary analytic methods.” (p. 11)

9. I was very much surprised by the finding that participants perceived AI as consistently more effortful. This finding almost seems like it should have served as a sanity check, since why would participants think AI would exert more effort than humans? I have read the authors’ discussion of this in the limitations section, but I remain unconvinced that it is not, as they also suggested, “a category error” resulting from how participants interpret the question. The authors’ alternative explanation for this is intriguing, and may point to something broader about the nature of human-AI relationship, which connects to their later discussion of these interactions. However, since the current study does not allow a definitive conclusion between the two proposed explanations, I recommend discussing this more cautiously and framing it as a question that requires further investigation in future research.

We agree that this finding is surprising and, as we acknowledge in the text, completely agree that the higher rating of effort for AI versus humans is unclear without understanding what exactly participants are attuning to when they make this rating. We have added a clause to the end of this paragraph to reiterate the lack of clarity in how we should interpret this finding:

“Still, this is speculative and future work needs to be done to explore what truly underlays this judgment of AI as effortful.” (p. 33)

10. Please refer to the supplementary analyses more clearly within the main text (e.g., by specifying the relevant pages and table/figure numbers). This would make the Supplementary Information easier to assess, as it is currently somewhat difficult to follow.

We appreciate this feedback and have done so throughout the main text.

Reviewer #3:

1. As stated by the authors (p. 6), the study builds on motivated empathy perspectives by adapting the empathy selection task. The motivated empathy perspectives are about the extent to which people feel empathy for others can be malleable and socially motivated. On the other hand, this study examines how much empathy people want to receive. That is, it takes the recipient-focused perspective rather than expresser- (empathizer-)focused perspective. It would be helpful to further elaborate on how the present findings relate to or extend the motivated empathy framework. For instance, could the motivation to

receive empathy be shaped by similar social and contextual factors that drive the motivation to feel empathy toward others?

Thank you for suggesting elaboration on this point. Motivated empathy accounts do traditionally take the empathizer's perspective (as most empathy research generally has up until this point), viewing empathy as a weighing of cost and benefits in deciding whether to select into an empathic situation. Motivated empathy reception applies in exactly the same way: empathy recipients can equally weigh costs and benefits before ultimately deciding whether to select into the *receipt* of empathy. We have added some additional detail to this paragraph acknowledging the motivated empathy perspective underlying the original Empathy Selection Task:

“Building on motivated empathy perspectives (Cameron et al., 2022; Keysers & Gazzola, 2014; Zaki 2014), we used a modified version of the empathy selection task (Cameron et al., 2019). This task views empathy as a motivated process in which individuals weigh its costs and benefits to ultimately make a decision to empathize. We adapted this general approach to examine motivated empathy *reception* rather than *expression*. Instead of presenting participants with situations in which they choose whether to *express* empathy for a target, participants could make the choice of which source to *receive* empathy from: human or AI.” (p. 7)

2. In the pilot study for Studies 1, 2, and 4 (generation of human and AI empathic responses), were the human and AI responses compared in terms of empathy, quality, and emotional reactions to confirm their equivalence? Ideally, individual raters should have

been randomly presented with both human- and AI-generated responses, without any labels indicating their source, and asked to rate them. Otherwise, it could be that the AI-generated responses were actually superior to the human-generated ones at baseline.

To clarify, we intentionally did not pre-test the stimuli to match them on empathy ratings. Such pre-testing would get at something closer to the effect of authorship label, as others have already studied (e.g., Yin et al., 2024; Rubin et al., 2025). These studies typically find that people rate empathy labelled as coming from an AI source as less empathetic than the same response labelled as human-generated. We believe, as prior work also suggests, that the AI generated responses were superior at baseline. Part of the interesting nature of the choice paradox effect is that, despite this superiority in empathic content, people still sought out human empathy. In other words, they seemed to weigh label more heavily than content. We have added the following paragraph to better convey the unique position of this relative to other studies that have experimented with the sole effect of author label on empathy ratings:

“We emphasize that, in this design, participants received responses that were actually created by the source of their choosing. That is, if they selected the ”HUMAN” deck, they received a human-created response and if they selected the “AI” deck, they received an AI-created response. Other studies have experimented with dissociating the true response author from the author label (e.g., labelling an AI-created response as human; Rubin et al., 2025; Yin et al., 2024). These studies typically find that people penalize AI sources of empathy, holding content constant. In contrast, we are interested in whether people

willingly prioritize empathy reception from humans or AI, while *fully knowing* the source. That is, we do not simply explore the separate effects of actual response authorship crossed with author label, but instead explore the unique question about empathy choice and the emotional consequences thereof, given accurate information of authorship. We believe this question is closer to real life empathy-seeking decisions as, naturalistically, people are likely to know that they are interacting with a human or a chatbot.” (pp. 7-8)

3. In real-world applications of generative AI, the AI agent is often anthropomorphized. For example, it is typically given a name (e.g., Alexia) rather than being labeled simply as “AI.” In contrast, the use of explicit labels (“human” vs. “AI”) in the present study may have accentuated the perceived differences between the two, reinforcing categorical distinctions and potentially fostering negative attitudes toward AI among those who hold less favorable attitudes toward algorithmic systems.

Thank you for raising this point. We completely agree that anthropomorphism could be relevant and, in our discussion section, acknowledge the tendency to anthropomorphize as a possibly relevant trait for empathy choice and additionally believe the possibility of AI voice features to be an extension of this anthropomorphism. While there are plenty of anthropomorphized AI assistant such as Alexa, Cortana, etc. we also would argue that one of the most commonly used chatbots, ChatGPT (which we used for this study), is non-anthropomorphized and simply labelled as an AI. For this reason, while we similarly think that anthropomorphism is a reasonable (and an extremely valuable and interesting) extension of this, we would also argue that because the most commonly used chatbot is

decidedly non-anthropomorphized, matching the general set up of our study, this is not a strong limitation. In sum, we certainly think it is possible that the non-anthropomorphism could accentuate negative attitudes for some participants, however, this accentuation may be quite naturalistic when we consider the form in which most people are interacting with LLMs currently.

We thank you again for your time, effort, and expertise in reviewing and helping us improve this manuscript.

Sincerely and on behalf of all authors,

Joshua Wenger, M.S.

Dear Reviewers,

We would, once again, like to express our appreciation for your thoughtful feedback regarding our manuscript now titled, “People choose to receive human empathy despite rating AI empathy higher” (COMMSPSYCHOL-25-0749-T), accepted in principle at *Communications Psychology*. We are grateful for your time and effort, and believe the insights you have provided have greatly improved our manuscript.

Below, we give point-by-point responses to feedback from each reviewer.

Reviewer #1:

Dear Editor,

I would like to thank you and the authors for the opportunity to review this timely and exceptionally well-crafted manuscript. The authors have thoroughly addressed all my concerns, including clarifying the conceptualisation of empathy, refining the description of methods and analyses, expanding the discussion to acknowledge alternative interpretations and limitations, and providing the generated empathic reactions for transparency. I believe these revisions have further strengthened the manuscript, and in my view, it is now ready for publication in its current form.

We thank the reviewer for their feedback. We agree that these revisions in response to their thoughtful suggestions have greatly strengthened the manuscript, and we sincerely appreciate the reviewer’s expertise, effort, and positive evaluation.

Reviewer #3:

All the points have been thoroughly addressed. Thank you for your careful revisions.

We thank the reviewer for their careful suggestions and their effort invested. We believe their constructive feedback has improved the final manuscript, and are pleased that we have addressed their points.

Once again, we thank you again for your guidance and investment of time and effort in reviewing and helping us improve this manuscript.

Sincerely and on behalf of all authors,

Joshua Wenger, M.S.